# Reducing Carbon Emissions for the Vehicle Routing Problem by Utilizing Multiple Depots

**Sihan Wang** [1] 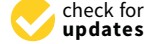**, Cheng Han** [1]**, Yang Yu** [1]**, Min Huang** [1]**, Wei Sun** [2,*] **and Ikou Kaku** [3]

1 State Key Laboratory of Synthetic Automation for Process Industries, College of Information Science and Engineering, Northeastern University, Shenyang 110819, China; wangsihan9@foxmail.com (S.W.); neu_hancheng@163.com (C.H.); yuyang@ise.neu.edu.cn (Y.Y.); mhuang@mail.neu.edu.cn (M.H.)
2 Business School, Liaoning University, Shenyang 110819, China
3 Department of Environmental Management, Tokyo City University, Yokohama 224-8551, Japan; kakuikou@tcu.ac.jp
* Correspondence: lnusunwei@163.com

**Abstract:** Emission reductions could be achieved by replacing the single-depot mode with a multi-depot mode of vehicle routing. In our study, we identified situations under which multiple depots could be used to effectively reduce carbon emissions. We proposed a branch-and-price (BAP) algorithm to obtain an optimal solution for the multi-depot green vehicle routing problem. Based on the BAP algorithm, we accurately quantified the carbon emission reduction potential of the multi-depot mode over the single-depot mode. Factors such as the number of depots, vehicle speed, customer demand, and service time were considered and analyzed. Computational tests were conducted, and the results showed that using multiple depots in a vehicle routing problem can reduce carbon emissions by at most 37.6%. In sensitivity analyses, we show relationships between these factors, and several managerial insights that can be used to successfully reduce carbon emissions were summarized.

**Keywords:** vehicle routing problem; multi-depot; carbon emission; sensitivity analyses

## 1. Introduction

Worldwide economic development has also led to significantly improved transportation and logistics services. However, in the past two decades, growth in the economy and logistics has caused a series of environmental problems, such as air pollution and climate abnormality. As the primary source of greenhouse gases, carbon emissions also play an important role in transportation and logistics. According to a report by the International Energy Agency [1], transportation is responsible for 24% of direct carbon emissions from fuel combustion. Meanwhile, road vehicles account for nearly three-quarters of transport carbon emissions, which makes vehicles in road transportation the main avenue for air pollution reduction.

In recent years, the vehicle routing problem (VRP) [2] that considers the reduction of carbon emissions or fuel consumption has attracted increasing attention [3,4]. This variant of the VRP is thus divided into two types: the green VRP (GVRP), which considers carbon emissions in routing and scheduling, and the pollution routing problem (PRP), which considers fuel consumption. Motivated by Chinese logistics development, our work considers how the multi-depot mode can maximize carbon emissions reduction. Driven by the Internet economy, the logistics industry in China has grown significantly, generating more than 10.3 trillion RMB in 2019. However, many problems remain unresolved. The efficiency of logistics in China is lower than that in developed countries. For example, the total cost of logistics in China was 14.6 trillion RMB in 2019, which was 14.7 percent of the GDP, while the corresponding rate in developed countries is 8–9%. On the other hand, the carbon emissions related to logistics accounts for 18.9% of the total carbon emissions in

China. Thus, it is worthwhile to investigate the relationship between energy consumption in logistics and carbon emissions.

Benefitting from the e-commerce boom, customers currently prefer to shop on the Internet. The purchased goods are delivered to buyers within several days by an express delivery service. Thus, many distribution centers have been established by logistics companies to store goods and deliver them to customers. Compared to the single-depot situation, multiple depots for logistics companies makes them more flexible and efficient in providing delivery services that also significantly reduce carbon emissions. In this paper, we study the multi-depot GVRP with time windows (MDGVRPTW) to reduce the total carbon emissions when scheduling vehicles for logistics services. We identify situations under which multiple depots can be used to effectively reduce carbon emissions. The use of a branch-and-price algorithm is proposed in this study to obtain the optimal solution of the multi-depot GVRPTW. Based on the algorithm, we accurately evaluate the effect of the multi-depot mode on the reduction of carbon emissions. Several main factors, such as the number of depots, vehicle speed, customer demands and service time, are considered and analyzed in this study. Given that pursuing minimum carbon emissions is the fundamental target of this work, it is inevitable that other vehicle routing costs have been overlooked.

The remainder of this paper is structured as follows. Section 2 presents a literature review of the MDGVRPTW. Section 3 provides the problem description and formulation for calculating carbon emissions, and a brief description of the branch-and-price algorithm is presented. Section 4 provides sensitivity analyses of the factors affecting carbon emissions in the MDGVRP. Section 5 provides a conclusion and some suggestions to efficiently reduce carbon emissions.

## 2. Literature Review

The traveling salesman problem (TSP) [5] is a classical problem that now has gained remarkable achievements. We refer readers to [5–8] and the references therein for the recent progress in these problems. As a generalization of the traveling salesman problem (TSP), the vehicle routing problem (VRP) has been greatly developed, and lots of articles about fractional calculus have been published. As the most practical and well-known variant of the VRP, the multi-depot vehicle routing problem (MDVRP) has been widely studied and utilized [9,10]. The MDVRP was first proposed by Tillman [11]. After many years of research, the MDVRP has been considered in many generalizations, and many exact, heuristic, and meta-heuristic algorithms have been developed to solve these generalizations. Typically, MDVRP models represent realistic cases, and heuristic algorithms are thus proposed to solve the large-scale MDVRPs. Zhou et al. [12] focus on the half open MDVRP with heterogeneous vehicles for hazardous materials transportation. Compared to the single-depot VRP, MDVRPs are more complicated for exact algorithms to solve; however, there are some exact algorithms for MDVRPs. For example, Bettinelli et al. [13] proposed a branch-and-cut-and-price algorithm for solving a multi-depot heterogeneous VRPTW. Contardo and Martinelli [14] proposed a column generation and cut algorithm to solve the MDVRP. Considering replenishment in scheduling, Muter et al. [15] solved the MDVRP using a branch-and-price algorithm. These multi-depot variants concentrate more on the total route cost of the problem.

Increasing amounts of carbon emissions in the atmosphere are gradually causing global climate change; thus, carbon emission reduction has become an effective measure to mitigate climate change. Işık et al. [16] considered the ecological footprint to evaluate the economic impact brought by carbon emissions and other types of pollution. Işık et al. [17,18] investigate the validity of the environment Kuznets curve hypothesis to effectively decrease the carbon emissions. In the VRP, there is a variant classified as the green vehicle routing problem (GVRP), which considers the effect of vehicle routing on the environment. Several reviews on green logistics and the GVRP have been presented to better understand the issues, such as [3,4,19–22]. To provide a more precise calculation of vehicle emissions, Demir et al. [23] analyzed several carbon emission models. Many studies have attempted to model

the GVRP by minimizing the total fuel consumption and operational cost, such as [24–28]. There have also been studies that directly considered carbon emissions as the main objective of the model. Wang et al. [29] proposed a cooperation strategy for green pickup and delivery problems to calculate the compensation and profit distribution. Peng et al. [30] solved the GVRP by using a population-based algorithm called memetic algorithm. Yu et al. [31] solved the heterogeneous fleet GVRP with a branch-and-price algorithm to minimize the total carbon emissions. Figliozzi [32] analyzed the efficiency of autonomous air and ground delivery vehicles to reduce carbon emissions. Pribyl et al. [33] considered using cooperative and automated vehicles and proposed a complementary approach that focuses on harmonizing the flow of traffic in urban centers to reduce carbon emissions. Saleh and Hatzopoulou [34] assessed the impact of private autonomous vehicles on greenhouse gas emissions. Zeng et al. [35] developed an eco-routing algorithm for navigation systems to find a path that consumes the minimum amount of gasoline. Wang and Wen [36] proposed an adaptive genetic algorithm to solve the two-echelon heterogeneous fleet GVRP. Li et al. [37] studied an electric VRP with constraints on battery life and battery swapping stations, an on hill climbing optimization and neighborhood search is developed to reduce carbon emissions and total logistics delivery costs.

Some MDGVRP studies have focused more on the algorithms used to solve the problem. Li et al. [38] considered shared depot resources for a MDGVRP, and the factors that affect the benefit ratio were analyzed. Li et al. [39] proposed an ant colony optimization algorithm to solve the MDGVRP with multiple objectives. Considering the time-dependent speed and piecewise penalty cost, Wang et al. [40] studied the MDGVRP with shared transportation resources, and they found that transportation resource sharing reduces travel distance and carbon emissions. Zhang et al. [41] considered using alternative fuel-powered vehicles in the MDGVRP to minimize the total carbon emissions. Peng et al. [42] presented a hybrid evolutionary algorithm to tackle the MDGVRP. Despite recent research on the MDGVRP, there is still a lack of analysis on the relationships between the related factors in the MDGVRP to reduce carbon emissions.

The scope of this study is to identify the MDGVRPTW, schedule vehicle routes to serve a set of customers, and minimize the total carbon emissions, subject to routing problem constraints, particularly load, speed, time windows, and the number of depots. A branch-and-price algorithm is proposed to optimize the potential carbon emissions reductions over multi-depot mode. Furthermore, sensitivity analyses for each factor in this problem were performed in order to explore the influence of carbon emissions reductions, such as the number of depots, vehicle speed, customer demands, and service time. Some useful suggestions that can efficiently reduce multi-depot carbon emissions are also provided.

## 3. Problem Description and Formulation

### 3.1. Mathematical Description of MDGVRPTW

Let $G = (V, A)$ be a completely directed multi-graph, where the set $V$ denotes the vertex set and $A$ denotes the arc set. $V = V_0 \cup V_d$, where $V_0 = \{1, 2, \dots, n\}$ denotes the set of customers, and $V_d = \{n + 1, n + 2, \dots, n + m\}$ represents the set of depots. Each customer $i \in V_0$ has a positive demand $f_i$, a service time $s_i$, and a time window $[e_i, l_i]$ that indicates the earliest and latest time the service starts at customer $i$. In this problem, if the vehicle reaches customer $i$ before time $e_i$, then it must wait until $e_i$. Additionally, the number of visits to each customer $i \in V_0$ is restricted to one. For the depots $i \in V_d$, let $f_i = e_i = s_i = 0$, and $l_i = H$, where $H$ represents the length of planning horizon. $A = \left\{ (i, j)^q : i, j \in V, q = 0, 1, \dots, Q \right\}$ represents the set of arcs, where $Q$ represents the maximal capacity of the vehicle. The arc $(i, j)^q$ indicates that a vehicle travels along arc $(i, j)$ with load $q$. A travel cost with carbon emissions $c_{ij}^q$, travel distance $d_{ij}$, and travel time $t_{ij}$ are associated with each arc $(i, j)^q \in A$. Several homogeneous fleets of vehicles with capacity $Q$ are located at different depots, and each depot has at most $K$ vehicles. These vehicles start their service from one depot and return to the original depot from which they left. In the MDGVRPTW, the optimal solution is composed of a set of feasible routes with minimal carbon emissions. A feasible route

corresponds to an elementary path $p = (i_0, i_1, i_2, \ldots, i_k, i_{k+1})$, $i_k \in V$ which consists of a subset of customers, where $i_0 = i_{k+1} \in V_d$. The route $p$ allows a maximum of $Q$ loads (i.e., $\sum_{j=1}^{k} f_{i_j} \leq Q$). The departure time $\tau_{i_j}$ at every visited vertex $i_j$ ($j \in \{0, 1, \ldots, k, k+1\}$) can be calculated recursively as follows.

$$\tau_{i_0} = e_{i_0} + s_{i_0} = 0, \tag{1}$$

$$\tau_{i_{j+1}} = \max\left\{e_{i_{j+1}} + s_{i_{j+1}}, \tau_{i_j} + t_{i_j i_{j+1}} + s_{i_{j+1}}\right\} \leq l_{i_{j+1}}, \forall j \in \{0, 1, \ldots, k\}. \tag{2}$$

Additionally, the carbon emissions of route $r$ are calculated by $c_r = \sum_{j=0}^{k} c_{i_j i_{j+1}}^{q_{i_j}}$, where $q_{i_j} = \sum_{t=j}^{k} f_{i_{t+1}}$ represents the total load when a vehicle travels along the arc $(i_j, i_{j+1})$. Consequently, the MDGVRPTW aims to obtain a set of feasible routes while minimizing the total carbon emissions.

### 3.2. Modeling Carbon Emissions

Several common carbon emissions calculation methods are reviewed by [21,23]. In this study, we adopt the method from Bektaş and Laporte [24]. The related parameters are listed in Appendix A, Table A1.

For an arc $(i, j)^q$, the carbon emissions when the vehicle leaves vertex $i$ to $j$ with loads $q$ can be calculated by

$$c_{ij}^q = \gamma \times \frac{\zeta}{\kappa}\left[(N_f N_e N_d + \frac{0.5 C_d \rho A v^3}{1000 \epsilon \overline{\omega}}) + \frac{v(g\sin\phi + gC_r\cos\phi + a_t)}{1000 \epsilon \omega} \times (w + q)\right] \times t_{ij} \tag{3}$$

The travel time $t_{ij} = d_{ij}/v$. Let $\alpha_1 = \gamma \times \frac{\zeta}{\kappa}\left(N_f N_e N_d + \frac{0.5 C_d \rho A v^2}{1000 \epsilon \overline{\omega}}\right)$, and $\alpha_2 = \gamma \times \frac{\zeta}{\kappa} \times \frac{(g\sin\phi + gC_r\cos\phi + a_t)}{1000 \epsilon \overline{\omega}}$; then, Equation (3) can be reformulated as follows:

$$c_{ij}^q = [\alpha_1 + \alpha_2(w + q)] \times d_{ij}. \tag{4}$$

Generally, for a feasible route $r$ with path $(i_0, i_1, \ldots, i_k, i_{k+1})$, $i_k \in V$, $i_0 = i_{k+1} \in V_d$, the carbon emissions can be represented as

$$c_r = \sum_{j=0}^{k} c_{i_j i_{j+1}}^{q_{i_j}} = (\alpha_1 + \alpha_2 w)\sum_{j=0}^{k} d_{i_j i_{j+1}} + \alpha_2 \sum_{j=0}^{k} f_{i_{j+1}} \sum_{t=0}^{j} d_{i_t i_{t+1}}. \tag{5}$$

### 3.3. Set-Partitioning Model for the MDGVRPTW

In this study, we propose a set-partitioning (SP) model for the MDGVRPTW. The SP model has been widely applied to solve VRPs with exact algorithms [13,15]. Let $\Omega_d$ denote the route set for all feasible routes of the depot $d \in V_d$. For each route $r$ in $\Omega_d$, a binary variable $\lambda_r$ is proposed to represent whether this route is used in the solution, and the carbon emissions for $r$ are represented by $c_r$. Let $a_{ir}$ be a 0-1 coefficient equal to 1 if vertex $i \in V$ is visited by route $r$, and 0 otherwise. Typically, the SP model for MDGVRPTW can be described as follows:

$$\min \sum_{d \in V_d} \sum_{r \in \Omega_d} c_r \lambda_r. \tag{6}$$

$$s.t. \sum_{d \in V_d} \sum_{r \in \Omega_d} a_{ir} \lambda_r = 1, \forall i \in V_c \tag{7}$$

$$\sum_{d \in V_d} \sum_{r \in \Omega_d} a_{ir} \lambda_r \leq K, \forall i \in V_d \tag{8}$$

$$\lambda_r \in \{0,1\}, \forall r \in \Omega_d, d \in V_d \tag{9}$$

In this SP model, the objective in (6) is to minimize the total carbon emissions. Constraint (7) limits each customer to only one visit, and constraint (8) ensures that every depot can schedule at most $K$ vehicles; in our study, the number of vehicles is not limited. Constraint (9) is the binary constraint. The relaxation of the SP model, which is called the master problem (MP), can be solved by proposing a column generation algorithm, and the routes with a negative reduced cost can be calculated by the pricing problem with a bidirectional labeling algorithm.

## 4. Methodology

In this section, we propose a branch-and-price (BAP) algorithm for solving the MDGVRPTW optimally, and the details are shown below. Combined with branch-and-bound and column generation, the BAP algorithm has become the most popular technology to obtain the optimal solution in VRPs. We refer readers to the paper by Costa et al. [43] for more information.

### 4.1. Column Generation Algorithm

To solve the relaxation of the SP model, we use the column generation algorithm; a simple description of this algorithm is shown in Algorithm 1 as follows:

---
**Algorithm 1:** Column generation algorithm for MDGVRPTW

---
**while** (**true**)
　　Solve the restricted MP (RMP) with added routes.
　　Obtain the dual prices corresponding to constraint (7, 8).
　　**for** ($d$ in $V_d$)
　　　Solve the bidirectional label-setting algorithm under distinct depot $d$.
　　**if** (No route with negative reduced cost exists)
　　　**break**
**else**
　　　Add the route with minimal reduced cost to RMP.
Output current solution in RMP as the final solution.
**return** final solution.

---

Let $\mu_i, i \in V_c$ and $v_d, d \in V_d$ denote the dual price of constraints (7, 8), respectively; then, the objective of the pricing problem can be described as follows:

$$\min\left\{c_r - \sum_{i \in V_c} a_{ir}\mu_i - v_d, r \in \Omega_d, d \in V_d\right\}. \tag{10}$$

From Equation (10), it can be seen that for each depot, the pricing problem must first calculate the corresponding route with minimal reduced cost. Then, at most $|V_d|$ routes with negative reduced costs need to be compared, and the minimal one is output to the SP model for calculation. In our study, we propose a bidirectional label-setting algorithm to solve the pricing problem.

### 4.2. Bidirectional Label-Setting Algorithm

In our study, we use a bidirectional label-setting algorithm reviewed by Costa et al. [43]. For a partial path $p_f$ in forward that starts at the depot $d$, we have that $p_f = \left\{i_1, \ldots, i_\updownarrow\right\}$, $i_\updownarrow \in V_0$, and let $v = i_{|p_f|}$ denote the latest visited customer. A label thus be presented to indicate the state of the partial path $p_f$ in the label-setting algorithm and can be described as a tuple $L_f = \left(p_f, d, v, \bar{c}, q, t, S\right)$. The reduced cost $\bar{c}$ associated with the partial path $\{d\} \cup p_f$ is presented, and can be calculated by $\bar{c} = \bar{c}_{d,i_1} + \sum_{\updownarrow=1:|p_f|} \bar{c}_{i_\updownarrow, i_{\updownarrow+1}}$, where $\bar{c}_{i,j}$ represents the route cost associated with dual variables of the arc $(i, j)$. In contrast to the general label-

setting algorithm in which the route cost is the distance of the arc, in the GVRPTW, the route cost is calculated by the multiple of the loads and distance, which is also described in Equation (4). The cumulative load $q$ is calculated by $q = \sum\limits_{i_{\updownarrow} \in p_f} f_{i_{\updownarrow}}$. The attribute $t$ in $L$ represents the earliest start time at customer $v$ for service, and the set of forbidden vertices $S$ that the vehicle cannot visit. A forward label can thus be initialized by the form $L_0 = (\{0\}, d, 0, 0, 0, 0, O)$. For an arc $(i, j)$ that $i = v(L_f)$ and $j \notin S$, let $L_f'$ denote the label that be extended by $L_f$ along the arc $(i, j)$, then the extension rule for forward labeling algorithm can thus be described as:

$$p(L_f') = p(L_f) \cup \{j\}, d(L_f') = d(L_f), v(L_f') = j, \bar{c}(L_f') = \bar{c}(L_f) + \bar{c}_{ij}, q(L_f') = q(L_f) + f_j,$$

$$t(L_f') = \min\left\{t(L_f) + t_{ij}, e_j\right\}, S(L_f') = S(L_f) \cup \left\{c \big| t(L_f') + t_{jc} > l_c \big| \big| q(L_f') + f_c > Q, c \in V_c\right\}.$$

For a partial path $p_b$ in backward that ends at depot $d$, we have that $p_b = \left\{i_{\updownarrow}, \ldots, i_1\right\}$, $i_{\updownarrow} \in V_0$. Then a tuple for backward label is presented by $L_b = (p_b, d, v, \bar{c}, q, t, S)$, where the difference with the forward label is that the attribute $t$ represents the latest start time at customer $v$ for service. Then the backward labeling algorithm is initialized by the form $L_0 = (\{0\}, d, 0, 0, 0, H, O)$. The extension rule for backward variant is similar to that of forward variant except for the arrival time and forbidden set for $L_b'$:

$$t(L_b') = \max\left\{t(L_b) - t_{ji}, l_j\right\}, S(L_b') = S(L_b) \cup \left\{c \big| t(L_b') - t_{cj} < e_c \big| \big| q(L_b') + f_c > Q, c \in V_c\right\}.$$

To eliminate those promising labels that cannot find the minimal reduced cost, dominance rules are thus proposed for both forward and backward labels. Let $L_1$ and $L_2$ denote separately two forward (backward) labels and $L_1$ dominates $L_2$ when following rules are satisfied: $d(L_1) = d(L_2), v(L_1) = v(L_2), \bar{c}(L_1) \leq \bar{c}(L_2), q(L_1) \leq q(L_2), t(L_1) \leq t(L_2)$ (for backward labels, $t(L_1) \geq t(L_2)$), and $S(L_1) \subseteq S(L_2)$.

After the extension for both forward and backward variants, a combination rule is utilized to concatenate both forward and backward labels. In our study, when the following rule is satisfied, then a forward and backward label can thus be concatenated: $v(L_f) = v(L_b)$, $t(L_f) \leq t(L_b), q(L_f) + q(L_b) \leq Q, S(L_f) \cap p(L_b) = \emptyset$.

### 4.3. Branching Rule

Two strategies are used in the BAP algorithm to obtain the lower and ultimate integer solutions of the MDGVRPTW. First, we branch on the number of vehicles used when the solution $\sum\limits_{d \in V_d} \sum\limits_{r \in \Omega_d} \bar{\lambda}_r$ is fractional. Then we branch on the arc if the number of vehicles used is an integer. This rule finds an arc in the routes constituting a fractional solution. If there is an arc $(i, j)$ whose fractional variable $\bar{\lambda}_r$ with a corresponding route $r$ is close to 0.5, then one sub-branch prohibits the arc $(i, j)$ when every feasible route leaves node $i$, and another branch prohibits any other arc $(i, k)$ $(k \in V_0 \backslash \{j\})$ after leaving node $i$ instead.

## 5. Discussion

In this section, we identify situations under which multiple depots can be used to effectively reduce carbon emissions. We aim to analyze the factors that can efficiently reduce the carbon emissions resulting from multiple depots. Several factors are proposed in the analysis, such as the number of depots, vehicle speed, customer demand, and service time. To reveal the effectiveness of our proposed branch-and-price method, we test the instances with 50 customers, and the details of these tests are shown in Table 1. Sensitivity analyses that compare different factors on carbon reduction are proposed and analyzed.

**Table 1.** Geographical distribution of depots.

| No. | Instance Class | 1 | 2 | 3 | 4 | 5 |
|---|---|---|---|---|---|---|
| | R | [35,35] | [0,0] | [67,77] | [0,77] | [67,0] |
| Coordinate | C | [40,50] | [22,25] | [75,58] | [13,63] | [65,20] |
| | RC | [40,50] | [0,0] | [75,58] | [14,73] | [70,20] |

### 5.1. Experiment Environment

The branch-and-price algorithm was coded in C# by embedding an ILOG CPLEX 12.10 solver to solve the relaxation of the set-partitioning model. The calculation time limit for each instance was set to 1 h. The results obtained when this limit was reached were considered to be the final solution of the instance.

The test cases used were derived from the Solomon benchmark instances (https://www.sintef.no/projectweb/top/vrptw/solomon-benchmark/ accessed date 29 December 2021) with three customer geographical distribution classes: R (random), C (clustered), and RC (random-clustered). Meanwhile, each class of instances is divided into two groups according to the length of time windows, those groups are denoted respectively by R1, R2, C1, C2, RC1, and RC2. Totally, 56 instances were used in the experiment.

For ease of computation, we modified the Solomon benchmark instances for a better analysis. In these instances, the time horizon was set at 24 h; thus, all customers' time windows can be scaled by a coefficient of $24/l_{n+1}$, where $l_{n+1}$ is the latest depot time. Moreover, the distance between every pair of vertices was set to $d_{ij}' = 2d_{ij}$.

### 5.2. Factor Setup

Factors such as the number of depots, vehicle speed, customer demand, and service time were considered in this comparison to estimate which factors affect the emission of carbon in vehicle routing. The setup and information for these factors are presented below:

Depot ($V_d$): number of depots. Let $D_i$ denote the number of depots. In this study, we set three values for $D_i$: $D_1 = 1$, $D_2 = 3$, and $D_3 = 5$. Moreover, based on the distribution of the customers, we set the geographical distribution of the depots, which are shown in Figure 1 for types R, C, and RC. The specific locations of the depots are listed in Table 1; when the number of depots was three, the data from the table for the first three columns were used in testing.

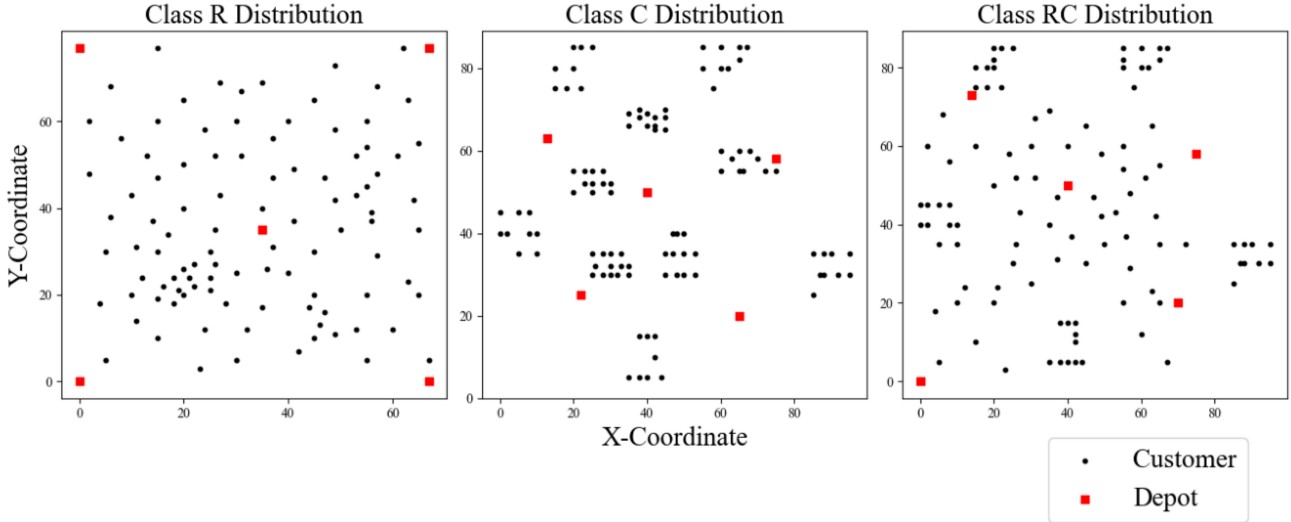

**Figure 1.** Geographical distribution of the customers and depots for distinct classes.

Vehicle speed ($v$). The vehicle speed is directly related to carbon emissions as given in Equation (3). Based on the report by (https://report.amap.com/download_city.do accessed date 29 December 2021) in China, the free flow speed in major cities in China is between 42 and 51 km/h, and we set our speed in this study at 42, 47, and 51 km/h, denoted by $v_1$, $v_2$, and $v_3$, respectively.

Customer demand ($F$). With the upsurge and development in e-commerce, logistics have also been actively developed. Increasing demand drives logistics companies to schedule more vehicles to satisfy customer requirements. On the other hand, with the rapid growth of GDP, differences in regional economic performances are increasingly evident.

The loads on a vehicle also affect the carbon emissions based on Equation (3); thus, the growth in demand is meaningful for evaluating how this change affects the carbon emission reduction. In our assumption, customers with larger demands have a higher requirement than general customers when they encounter aggregate demand growth. To analyze how aggregate demand growth affects carbon emissions under the mode of multiple depots, we created a new case for this situation. The sum of demands for a problem can be represented as $sf = \sum\limits_{i \in V_c} f_i$, and the average demands are $avg = \frac{sf}{n}$, where $n$ denotes the number of customers. Let the rate of growth of the aggregate demand increase by 20%, then for each customer $l$ whose demands satisfy the condition $f_l \geq avg$, the increased demands $f_l'$ of this customer can be calculated by

$$f_l' = f_l + \left\lfloor \frac{0.8 \times 0.2 \times sf \times f_l}{\sum\limits_{i \in V_c \& f_i \geq avg} f_i} \right\rfloor \tag{11}$$

where $0.2 \times sf$ represents the growth of the total demands, and customers whose demands exceed $avg$ would hold 80% of increasement (i.e., $0.8 \times 0.2 \times sf$). On the other hand, for customer $l$ whose $f_l < avg$, the new demand $f_l'$ is thus calculated by:

$$f_l' = f_l + \left\lfloor \frac{0.2 \times 0.2 \times sf \times f_l}{\sum\limits_{i \in V_c \& f_i < avg} f_i} \right\rfloor \tag{12}$$

Therefore, let $F_1$ denote instances with unchanged demands and $F_2$ denote instances with increased demands.

Service time ($t$). Generally, the service time for each customer in this study was fixed at 0.5 h. However, in reality, the service time is more relevant to the goods delivered by vehicles. Therefore, service time based on customers' demands was proposed to test the instance. Let $T_1$ denote the fixed service time for every customer, $T_2$ denote the demands-based service time for every customer, and the new service time $s_i'$ for customer $i$ can be calculated by:

$$s_i' = \frac{0.5 \times f_i}{\max\{f_l, l \in V_c\}} \tag{13}$$

In total, there were six parameters with 16 conditions, the details of which are listed in Table 2. Moreover, 216 ($3 \times 2 \times 3 \times 3 \times 2 \times 2$) combinations were examined in this study for various conditions of each parameter. In this study, 2016 ($56 \times 3 \times 3 \times 2 \times 2$) tests were executed, and the details are described in Section 5.4. For different parameter conditions of the tested instance, we used the following naming convention: <class>–$D$–$v$–$F$–$T$. For example, R1–$D_1$–$v_1$–$F_1$–$T_1$ denotes an instance of group R1 with conditions $D_1$, $v_1$, $F_1$, and $T_1$ for the different parameters.

**Table 2.** Classification of the factors.

| Factor | Class of Instances | Number of Depots | Speed | Demands | Service Time |
|---|---|---|---|---|---|
| | R1 | $D_1$ | $v_1$ | $F_1$ | $T_1$ |
| | R2 | $D_2$ | $v_2$ | $F_2$ | $T_2$ |
| Condition | C1 | $D_3$ | $v_3$ | | |
| | C2 | | | | |
| | RC1 | | | | |
| | RC2 | | | | |

### 5.3. Performance of the BAP Algorithm

The BAP is time-consuming and cannot obtain the optimal solution within the time limit, and the result obtained at the time limit still has a small gap with the relaxation of the model at the root node. This feature of the BAP makes it relatively stable to the optimal solution, and its results are suitable for our analysis. In total, the branch-and-price algorithm in this study is executed within an hour limit, and once the calculation is terminated, the current upper bound of the solution is the final solution of the solved instance.

In this study, the number of customers is set at 50, and the details of the solved instances are listed in Tables A2–A4 in Appendix A. The column "$^{\#}$Root" denotes the relaxation solution at root node, column "$^{\#}$Opt" denotes the best-obtained solution obtained by the BAP with 1 h limit, and the "Gap (%)" represents the gap between the current lower bound obtained at the time when ends\terminates of the algorithm to the solved instance and the best-obtained solution. The brief description for the performance of the BAP is shown in Table 3, where the column "No. of inst" represents the total number of instances, columns under factors (i.e., $D_1$, $D_2$, and $D_3$) represent the number of solved instances, average computing time of solved instances, and average gap of instances between the lower bound and the upper bound when the algorithm is terminated.

**Table 3.** Performance of the BAP algorithm.

| Class | No. of Inst | $D_1$ | | | $D_2$ | | | $D_3$ | | |
|---|---|---|---|---|---|---|---|---|---|---|
| | | No. of Solved | Time (s) | Gap (%) | No. of Solved | Time (s) | Gap (%) | No. of Solved | Time (s) | Gap (%) |
| R1 | 12 | 12 | 1343.3 | 0 | 12 | 337.31 | 0 | 12 | 168.83 | 0 |
| R2 | 11 | 11 | 1408.28 | 0 | 11 | 369.43 | 0 | 11 | 136.21 | 0 |
| C1 | 9 | 8 | 914.04 | 0.03 | 5 | 319.95 | 0.71 | 4 | 586.65 | 1.27 |
| C2 | 8 | 8 | 95.61 | 0 | 6 | 1005.8 | 0.04 | 8 | 1194.13 | 0 |
| RC1 | 8 | 1 | 2862.65 | 6.08 | 1 | 102.07 | 5.24 | 1 | 84.55 | 5.1 |
| RC2 | 8 | 1 | 10.42 | 5.91 | 1 | 7.17 | 4.68 | 2 | 330.26 | 3.89 |
| Average | | | 1105.72 | 2 | | 356.96 | 1.78 | | 416.77 | 1.71 |

Table 3 illustrates that the BAP algorithm can obtain relatively good results in instances in all classes, and the average gaps of these three are less than 2%. The results also illustrate that most instances in classes RC1 and RC2 cannot be solved in optimality within the time limit, but the average gap is relatively small. Though the BAP algorithm is time-consuming when searching for appropriate optimal results, it still can be utilized to get optimal\suboptimal results within one hour limit. Consequently, the BAP algorithm can be seen as relatively stable and sufficiently competitive for our comparison and analysis.

### 5.4. Sensitivity Analyses

In this section, we identify situations under which multiple depots can be utilized to effectively reduce carbon emissions. First, we compare the impact on carbon emissions when changing a single condition for a single factor in the problem. The baseline of the carbon emissions is provided by condition $*-D_1-v_1-F_1-T_1$, where the asterisk (*) represents an arbitrary group of the Solomon instances.

The average reduction in carbon emissions for each group is described in Table 4; the conditions $D_1$, $v_1$, $F_1$, and $T_1$ are not listed in the table because they represent the conditions in the baseline problem solution. For each instance in the Solomon group, the reduction in carbon emissions can be evaluated using the formula $\frac{reduced\,carbon\,emissions}{original\,carbon\,emissions} \times 100\%$.

**Table 4.** Average reduction of carbon emissions (%) from a single condition change.

| Condition\Group | R1 | R2 | C1 | C2 | RC1 | RC2 | Avg. R (%) |
|---|---|---|---|---|---|---|---|
| $D_2$ | 4.8 | 4.7 | 11.4 | 9.3 | 14 | 14 | 9.7 |
| $D_3$ | 11.6 | 11.7 | 23.3 | 26.4 | 32.7 | 32.8 | 23.1 |
| $v_2$ | 4.9 | 4.7 | 4.3 | 4.4 | 4.3 | 4.4 | 4.5 |
| $v_3$ | 7.4 | 6.5 | 6.1 | 5.8 | 6.5 | 6.6 | 6.5 |
| $F_2$ | −4.7 | −5.1 | −14.6 | −12.3 | −1 | −1.2 | −6.5 |
| $T_2$ | 2.8 | 1.4 | 0.4 | 0 | 0.9 | 0.8 | 1.1 |

Figure 2 also depicts the comparison of the carbon emissions reductions with changed conditions and corresponds to Table 4, where the column "Avg. R (%)" represents the average reduction of an instance among the Solomon instance groups. Table 4 illustrates that a change in condition $D_3$ significantly reduced carbon emissions by an average of 23.1%. Compared to condition $D_2$, the reduction in carbon emissions for $D_3$ was more than double. Most importantly, the multi-depot mode in group RC showed a superior ability to reduce carbon emissions. Changes in vehicle speed decreased carbon emissions by 4.5% and 6.5%, which did not indicate a superior performance. Additionally, the difference in scheduling horizon (for example, R1 and R2) was less crucial for reducing carbon emissions, even when considering the load-based service time ($T_2$); a reduction of only 1.1% was obtained compared to the original problem. On the other hand, the growth in customer demand increased carbon emissions by 6.5%, especially when the distribution of customers was cluster-based (i.e., groups C1 and C2)

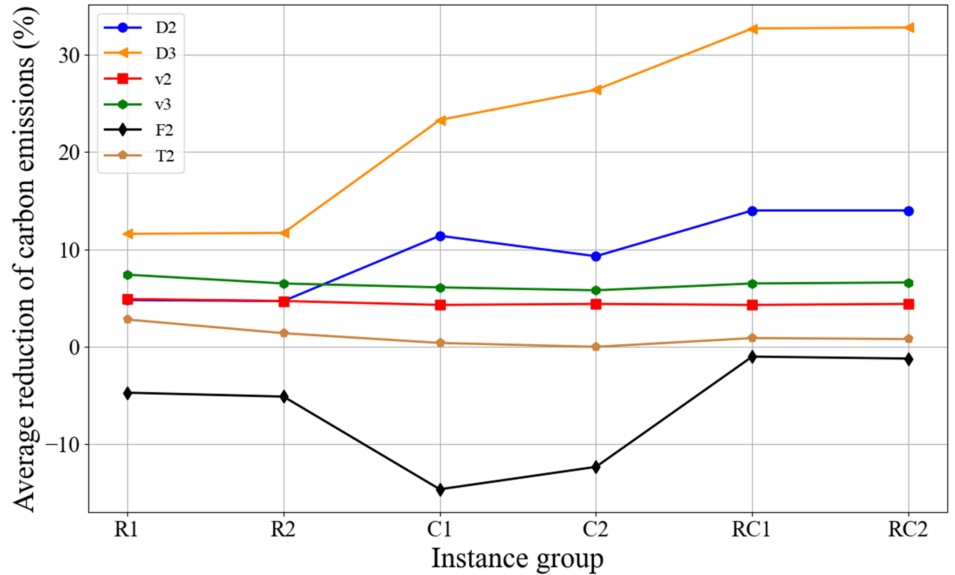

**Figure 2.** Average reduction in carbon emissions for a single condition change.

Table A5 in Appendix A indicates the reduction in carbon emissions for different combinations of conditions. For example, the average reduction in groups *–$D_2$–$v_1$–$F_1$–$T_1$ was 9.7 percent. To reveal the interrelationship between the multi-depot condition and other conditions, the influence of each factor on carbon emission reductions is analyzed in detail as follows.

Depot ($V_d$). The number of depots significantly affects the reduction in carbon emissions, and the results of the average reduction for conditions $D_1$ to $D_3$ are depicted in Figure 3. Compared to the other conditions, $D_3$ showed the most reduced carbon emissions (at most a 37.6% reduction) for the vehicle routing under all conditions, and the reduction in $D_3$ was twice that in $D_2$. Moreover, under the condition $D_3$, customer distribution significantly affected the carbon emissions reductions. From Figure 2, it can be seen that

the advantage of $D_3$ in groups R1 and R2 was small and occurred only when the customer distribution was cluster related (i.e., groups C and RC). Thus, if customer distribution is clustered, multiple depots can significantly reduce the carbon emissions. In our test, the best reduction in carbon emissions was obtained under conditions of $D_3$–$v_3$–$F_1$–$T_2$, and undoubtedly, groups RC1 and RC2 benefited most from these conditions. These two groups showed average carbon emissions reductions of 37.5% compared to the original conditions with a single depot.

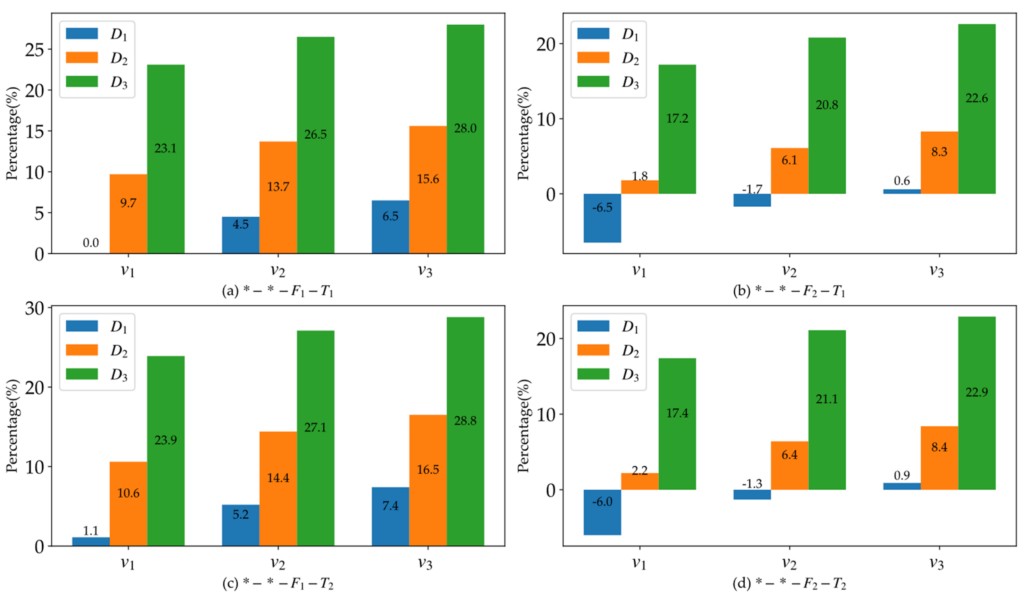

**Figure 3.** Reduction in carbon emissions with different numbers of depots, where the first asterisk (*) represents the condition of the number of depots and the second asterisk represents the condition of vehicle speed.

Vehicle speed ($v$), From Figure 4 it can be observed that the carbon emissions reductions caused by vehicle speed were relatively weak. At the same speed, the reductions for different distributions were close, which indicates that customer distribution has little effect on carbon emissions reductions when the vehicle speed changes. As shown in Table A5, changes in vehicle speed under different conditions caused a stable reduction in carbon emissions. We then analyzed the effect of the rate of reduction between conditions $v_1$ to $v_2$ and $v_1$ to $v_3$. The difference was calculated using column "Avg. R (%)" in Table A5 and depicted in Figure 4. The figure shows that the increased vehicle speed influences the carbon emissions for the single depot condition more, although the multi-depot condition almost always causes a reduction in carbon emissions. From Figure 4, the maximum reduction provided by vehicle speed was under conditions $D_1$–$v_3$–$F_2$–$T_1$ at a rate of 7.1%. Consequently, improving the vehicle speed benefits the carbon emissions reductions more when there is only one depot. When there are multiple depots, carbon emissions can be reduced to a smaller extent by increasing vehicle speed.

Customer demand ($F$). There is no doubt that the growth in customer demand leads to an increase in carbon emissions, especially when the customer distribution is C1 and C2 (see instances $D_1$–*–$F_2$–* in Table A5). On the other hand, multiple depots can not only significantly cover the customer requirements but also reduce total carbon emissions. To a certain extent, the increase in carbon emissions provided by the growing requirements of customers can be counteracted by multiple depots. In Table A5 in Appendix A, the growth in customer demands under the single-depot condition increased carbon emissions by a maximum of 14.5% (see conditions C1–$D_1$–$v_1$–$F_2$–$T_2$); however, under multi-depot conditions, the vehicles can save 32.8% of the emissions (see conditions C2–$D_3$–$v_1$–$F_2$–$T_2$ and C2–$D_1$–$v_1$–$F_2$–$T_2$).

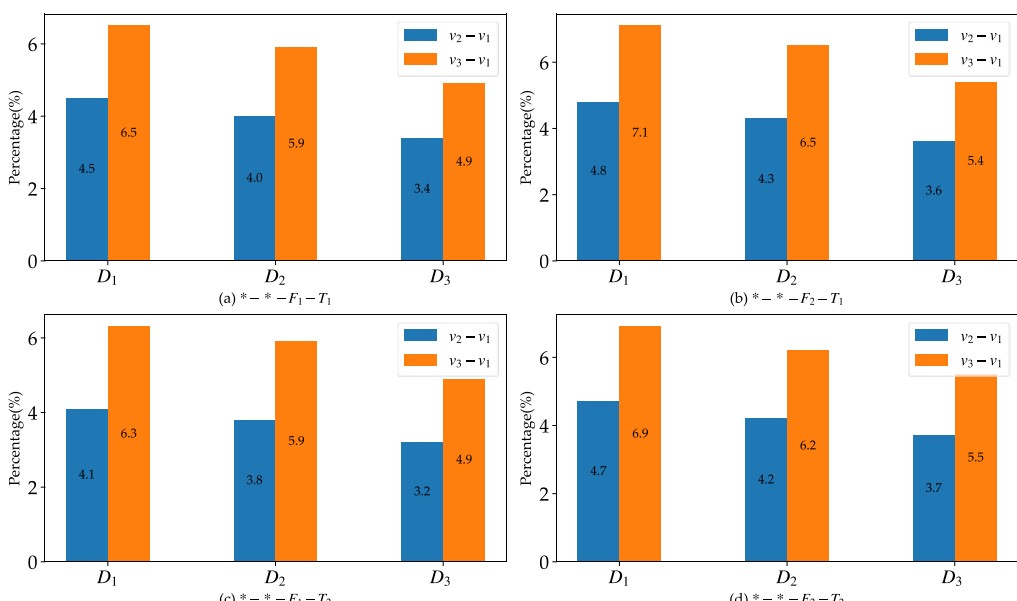

**Figure 4.** Difference in reduction rates for different vehicle speeds, where the first asterisk (*) represents the condition of the number of depots and the second asterisk represents the condition of vehicle speed.

Service time ($t$). Changes in the service time do not significantly affect carbon emissions reductions. The results in Table A5 demonstrate that more flexible service time to customers can reduce total carbon emissions, but the impact is limited. To improve the comparison, Table A6 in Appendix A compares the results when the service time is changed from $T_1$ to $T_2$. It can be seen that the demand-based service time can reduce carbon emissions more when the customer distribution is a type R (see R1 and R2). Moreover, when multiple depots are applied, the reduction obtained from $T_2$ instantly becomes irrelevant, with an average reduction of 0.5% in the multi-depot mode.

Overall, carbon emissions reductions provided by multiple depots is the most useful and straightforward approach when other costs are not considered. From the preceding analysis, we demonstrate that increasing the number of depots and vehicle speed is effective in reducing carbon emissions. When facing customer demand growth, increasing the number of depots is an efficient way to both cover customer demands and reduce carbon emissions. Based on this study, changing the service time for each customer according to their demands can also reduce carbon emissions, although the reduction is small.

## 6. Conclusions

In this study, we identified conditions under which multiple depots can be utilized to effectively reduce carbon emissions. A branch-and-price algorithm was proposed to obtain the optimal solution of the instance. Based on the results, we accurately evaluated the carbon emissions reductions potential for the multi-depot mode over the single-depot mode. Numerous experiments were conducted, and a maximum reduction of 37.6% in carbon emissions was achieved using the multi-depot mode with other factors. We also analyzed the influence of vehicle speed, customer demand, and service time on the carbon emissions reductions caused by multiple depots. Different customer distributions also affected the reduction in carbon emissions. Consequently, we propose the following recommendations for effective measures in carbon emissions reductions:

- Without considering other costs, the multi-depot mode is the most useful and beneficial way to reduce carbon emissions, especially when the customer distribution is RC (semi-clustered). Based on our tests, carbon emissions can be effectively reduced by a maximum of 37.6%.

- The single-depot mode benefits more from improving the vehicle speed to reduce carbon emissions, and at most a 7.1% reduction can be achieved by changing the vehicle speed. On the other hand, improving the speed of vehicles is the most direct method of reducing carbon emissions without changing other factors.
- The growth in customer requirements could cause more greenhouse gases to be emitted into the environment. However, this type of growth in carbon emissions can be counteracted somewhat by multiple depots.
- The service time to customers has little effect on carbon emissions, especially when multiple depots are utilized.

Limited to the current crowd traffic condition, it seems hard to increase the vehicle speed to reduce carbon emissions. Companies should provide their service during off-peak periods to not only reduce carbon emissions but also improve operational efficiency. On the other hand, carbon emissions reduction could be further benefited by appropriate depots, this will definitely be a long-term benefit for both companies and the environment.

Since the data used in this study comes from the Solomon benchmark instances, more practical instances would be considered in future research. Moreover, the growing popularity of hybrid and electric vehicles has spurred a lot of research into greener, more efficient, more sustainable, and less environmentally damaging transportation. In future research, we will analyze the benefits of using hybrid and electric vehicles with multiple rechargeable depots.

**Author Contributions:** Conceptualization, Y.Y.; Data curation, S.W.; Formal analysis, S.W. and C.H.; Funding acquisition, W.S.; Methodology, Y.Y.; Software, C.H.; Supervision, M.H.; Writing—original draft, S.W.; Writing—review & editing, W.S. and I.K. All authors have read and agreed to the published version of the manuscript.

**Funding:** This research was funded by the National Natural Science Foundation of China, grant number 71831006, 72171043, and 71620107003.

**Institutional Review Board Statement:** Not applicable.

**Informed Consent Statement:** Not applicable.

**Data Availability Statement:** Not applicable.

**Conflicts of Interest:** The authors declare no conflict of interest.

## Appendix A

**Table A1.** Typical vehicle parameter values.

| Notation | Description | Typical Values |
|---|---|---|
| $Q$ | Capacity (Kg) | 1200 |
| $w$ | Weight of empty vehicle (Kg) | 1890 |
| $\rho$ | Air density (kg/m$^3$) | 1.2041 |
| $A$ | Frontal surface of the vehicle (m$^2$) | 4 |
| $g$ | Gravitational constant (m/s$^2$) | 9.81 |
| $\zeta$ | Fuel-to-air mass | 1 |
| $\phi$ | Declination of the road | 0 |
| $C_d$ | Coefficient of aerodynamic drag | 0.7 |
| $C_r$ | Rolling resistance | 0.01 |
| $v$ | Vehicle velocity (m/s) | 11.67 (42 km/h) |
| $a_t$ | Acceleration (m/s$^2$) | 0 |
| $K$ | Heating value of typical diesel fuel (kJ/g) | 44 |
| $N_f$ | Engine friction factor (kJ/rev/liter) | 0.2 |
| $N_e$ | Engine speed (rev/s) | 40 |
| $N_d$ | Engine displacement (liters) | 5 |
| $\epsilon$ | Vehicle drive train efficiency | 0.4 |
| $\overline{\omega}$ | Efficiency parameter for diesel engines | 0.9 |
| $\gamma$ | Index of fuel to carbon emission | 3.164 |

**Table A2.** Computational results of solved instances for the one-depot mode.

| Inst | #Root | #Opt | Gap (%) | Time | Inst | #Root | #Opt | Gap (%) | Time |
|------|-------|------|---------|------|------|-------|------|---------|------|
| r101 | 619.29 | 621.63 | 0 | 6.06 | c106 | 333.74 | 357.34 | 0 | 73.25 |
| r102 | 533.2 | 538.51 | 0 | 57.55 | c107 | 332.09 | 354.81 | 0 | 186.84 |
| r103 | 471.12 | 479.95 | 0 | 157.97 | c108 | 331.08 | 351.68 | 0 | 68.57 |
| r105 | 544.62 | 545.92 | 0 | 24.66 | c201 | 415.62 | 429.39 | 0 | 16.78 |
| r106 | 491.88 | 498.01 | 0 | 145.28 | c202 | 413.03 | 426.29 | 0 | 42.78 |
| r107 | 449.16 | 461.88 | 0 | 1098.33 | c203 | 407.62 | 418.69 | 0 | 217.94 |
| r108 | 421.76 | 432.3 | 0 | 2184.29 | c204 | 405.75 | 416.07 | 0 | 356.88 |
| r109 | 467.48 | 480.71 | 0 | 1057.52 | c205 | 412.02 | 417.33 | 0 | 4.09 |
| r110 | 438.94 | 455.9 | 0 | 2568.15 | c206 | 411.38 | 420.75 | 0 | 25.43 |
| r111 | 441.47 | 451.95 | 0 | 1214.95 | c207 | 409.74 | 416.6 | 0 | 63.51 |
| r112 | 420.34 | 431.22 | 0 | 1260.88 | c208 | 410.85 | 417.3 | 0 | 37.47 |
| r201 | 568.16 | 574.04 | 0 | 41.44 | rc101 | 575.03 | 641.85 | 0 | 2862.65 |
| r202 | 509.85 | 517.28 | 0 | 215.62 | rc102 | 557.88 | 632.61 | 6.5 | - |
| r203 | 457.64 | 465.23 | 0 | 56.63 | rc103 | 544.56 | 623.34 | 7.3 | - |
| r204 | 426.48 | 439.92 | 0 | 10050.5 | rc104 | 528.58 | 613.63 | 7.6 | - |
| r205 | 499.6 | 505.13 | 0 | 9.54 | rc105 | 560.5 | 634.26 | 6.3 | - |
| r206 | 465.84 | 472.64 | 0 | 70.96 | rc106 | 544.24 | 627.64 | 6.7 | - |
| r207 | 437.52 | 444.95 | 0 | 53.06 | rc107 | 529.94 | 614.67 | 6.7 | - |
| r208 | 421.13 | 431.82 | 0 | 594.92 | rc108 | 526.77 | 612.87 | 7.5 | - |
| r209 | 458.34 | 471.44 | 0 | 456.34 | rc201 | 601.16 | 657.51 | 0 | 10.42 |
| r210 | 473.27 | 485.67 | 0 | 630.78 | rc202 | 569.33 | 643.53 | 6.2 | - |
| r211 | 431.79 | 446.73 | 0 | 3311.23 | rc203 | 545.63 | 623.4 | 7.2 | - |
| c101 | 333.74 | 358.23 | 0 | 41.24 | rc204 | 528.58 | 618.69 | 8.3 | - |
| c102 | 331.14 | 350.98 | 0 | 608.34 | rc205 | 572.11 | 642.7 | 5.5 | - |
| c103 | 328.87 | 341.78 | 0 | 3455.21 | rc206 | 558.95 | 633.72 | 5.2 | - |
| c104 | 326.24 | 345.89 | 0.3 | - | rc207 | 539.37 | 625.4 | 6.9 | - |
| c105 | 333.3 | 355.19 | 0 | 54.46 | rc208 | 527 | 616.66 | 8 | - |

**Table A3.** Computational results of solved instances for the three-depot mode.

| Inst | #Root | #Opt | Gap (%) | Time | Inst | #Root | #Opt | Gap (%) | Time |
|------|-------|------|---------|------|------|-------|------|---------|------|
| r101 | 582.78 | 587.02 | 0 | 6.76 | c106 | 298.49 | 316.21 | 0 | 37.11 |
| r102 | 507.82 | 510.61 | 0 | 77.11 | c107 | 295.39 | 313.2 | 0 | 578.21 |
| r103 | 453.21 | 459.13 | 0 | 99.28 | c108 | 292.73 | 310.01 | 1 | - |
| r104 | 412.9 | 422.36 | 0 | 299.61 | c109 | 289.47 | 308.93 | 2 | - |
| r105 | 510.64 | 513.55 | 0 | 6.7 | c201 | 376.48 | 382.65 | 0 | 14.25 |
| r106 | 462.23 | 469.22 | 0 | 22.13 | c202 | 373.42 | 382.65 | 0 | 320.51 |
| r107 | 431.76 | 441.38 | 0 | 112.85 | c203 | 368.99 | 378.22 | 0 | 1397.41 |
| r108 | 402.78 | 414.55 | 0 | 1945.46 | c204 | 363.45 | 372.66 | 0.2 | - |
| r109 | 449.89 | 460.36 | 0 | 202.54 | c205 | 374.3 | 381.96 | 0 | 131.24 |
| r110 | 417.46 | 428.28 | 0 | 337.39 | c206 | 373.35 | 382.94 | 0 | 409.82 |
| r111 | 422.36 | 426.76 | 0 | 858.39 | c207 | 368.89 | 380.22 | 0.1 | - |
| r112 | 399.77 | 408.82 | 0 | 79.52 | c208 | 373.26 | 381.61 | 0 | 3761.58 |
| r201 | 543.26 | 545.41 | 0 | 6.12 | rc101 | 511.65 | 562.53 | 0 | 102.07 |
| r202 | 488.71 | 494.15 | 0 | 64.8 | rc102 | 492.6 | 547.69 | 4.8 | - |
| r203 | 443.09 | 450.11 | 0 | 187.42 | rc103 | 481.88 | 540.44 | 6.1 | - |
| r204 | 408.66 | 414.83 | 0 | 1453.08 | rc104 | 466.18 | 526.81 | 6.8 | - |
| r205 | 485.93 | 490.96 | 0 | 76.76 | rc105 | 496.12 | 547.88 | 4.6 | - |
| r206 | 444.42 | 449.89 | 0 | 142.9 | rc106 | 481.81 | 538.73 | 5.8 | - |
| r207 | 420.37 | 428.6 | 0 | 436.83 | rc107 | 466.88 | 528.23 | 6.7 | - |
| r208 | 400.37 | 409.86 | 0 | 519.95 | rc108 | 464.32 | 526.58 | 7.1 | - |
| r209 | 436.66 | 450.21 | 0 | 371 | rc201 | 531.27 | 565.91 | 0 | 7.17 |

**Table A3.** *Cont.*

| Inst | #Root | #Opt | Gap (%) | Time | Inst | #Root | #Opt | Gap (%) | Time |
|------|-------|------|---------|------|------|-------|------|---------|------|
| r210 | 451.41 | 462.61 | 0 | 460.02 | rc202 | 497.67 | 549.49 | 4.5 | - |
| r211 | 408.54 | 418.76 | 0 | 344.8 | rc203 | 482.54 | 540.44 | 6.1 | - |
| c101 | 298.87 | 315.83 | 0 | 31.64 | rc204 | 466.53 | 526.81 | 6.6 | - |
| c102 | 292.8 | 312.26 | 0 | 912.02 | rc205 | 504.53 | 550.79 | 3 | - |
| c103 | 286.97 | 306.53 | 0.7 | - | rc206 | 498.52 | 547.25 | 4.3 | - |
| c104 | 281.41 | 307.19 | 2.7 | - | rc207 | 477.73 | 533.32 | 5.8 | - |

**Table A4.** Computational results of solved instances for the five-depot mode.

| Inst | #Root | #Opt | Gap (%) | Time | Inst | #Root | #Opt | Gap (%) | Time |
|------|-------|------|---------|------|------|-------|------|---------|------|
| r101 | 539.02 | 541.61 | 0 | 7.61 | c106 | 260.21 | 275.17 | 0 | 50.06 |
| r102 | 473.02 | 468.02 | 0 | 12.53 | c107 | 255.02 | 264.13 | 0 | 1452.71 |
| r103 | 417.14 | 418.63 | 0 | 35.93 | c108 | 252.27 | 266.35 | 0.7 | - |
| r104 | 381.99 | 383.95 | 0 | 298.49 | c109 | 249.67 | 263.47 | 1.3 | - |
| r105 | 478.14 | 478.14 | 0 | 3.52 | c201 | 304.57 | 318.2 | 0 | 17.61 |
| r106 | 433.97 | 433.97 | 0 | 6.8 | c202 | 305.23 | 311.62 | 0 | 165.52 |
| r107 | 405.69 | 407.49 | 0 | 116.02 | c203 | 299.92 | 309.91 | 0 | 4126.34 |
| r108 | 377.21 | 376.99 | 0 | 748.11 | c204 | 295.9 | 299.98 | 0 | 1003.86 |
| r109 | 433.94 | 439.73 | 0 | 260.2 | c205 | 305.07 | 310.26 | 0 | 60.42 |
| r110 | 396.01 | 402.43 | 0 | 139.63 | c206 | 299.6 | 309.12 | 0 | 111.74 |
| r111 | 399.68 | 404.8 | 0 | 210.78 | c207 | 296.97 | 305.29 | 0 | 3025.16 |
| r112 | 374.49 | 378.51 | 0 | 186.33 | c208 | 299.63 | 305.61 | 0 | 1042.35 |
| r201 | 498.87 | 498.87 | 0 | 2.64 | rc101 | 407.78 | 444.92 | 0 | 84.55 |
| r202 | 460.05 | 461.44 | 0 | 24 | rc102 | 392.11 | 433.18 | 5.6 | - |
| r203 | 408.57 | 408.79 | 0 | 21.93 | rc103 | 382.91 | 420.91 | 5.2 | - |
| r204 | 379.49 | 384.96 | 0 | 347.49 | rc104 | 369.11 | 410.31 | 7.6 | - |
| r205 | 464.48 | 466.34 | 0 | 20.89 | rc105 | 396.54 | 432.27 | 3.7 | - |
| r206 | 420.75 | 423.88 | 0 | 56.49 | rc106 | 378.07 | 415.46 | 5.2 | - |
| r207 | 394.9 | 396.77 | 0 | 66.31 | rc107 | 371.11 | 408.31 | 6.2 | - |
| r208 | 375.03 | 380.19 | 0 | 636.76 | rc108 | 367.37 | 405.56 | 7.3 | - |
| r209 | 419.14 | 422.96 | 0 | 11.88 | rc201 | 418.12 | 449.29 | 0 | 30.22 |
| r210 | 419.51 | 425.24 | 0 | 225.15 | rc202 | 397.94 | 434.96 | 4 | - |
| r211 | 384.17 | 389.77 | 0 | 84.78 | rc203 | 389.65 | 419.17 | 4.7 | - |
| c101 | 260.59 | 276.72 | 0 | 28.18 | rc204 | 369.11 | 407.9 | 7.1 | - |
| c102 | 254.32 | 271.95 | 1.8 | - | rc205 | 407.97 | 438.02 | 0 | 630.3 |
| c103 | 245.91 | 263.21 | 1.8 | - | rc206 | 397.08 | 427.58 | 2.1 | - |
| c104 | 240.05 | 267.48 | 5.8 | - | rc207 | 379.87 | 416.19 | 5.6 | - |

**Table A5.** Average reduction of carbon emissions (%) for combination of multiple conditions.

| Condition | $D_1$ | | | | | | Avg. R (%) | $D_2$ | | | | | | Avg. R (%) | $D_3$ | | | | | | Avg. R (%) |
|---|---|---|---|---|---|---|---|---|---|---|---|---|---|---|---|---|---|---|---|---|---|
| | R1 | R2 | C1 | C2 | RC1 | RC2 | | R1 | R2 | C1 | C2 | RC1 | RC2 | | R1 | R2 | C1 | C2 | RC1 | RC2 | |
| $v_1-F_1-T_1$ | 0 | 0 | 0 | 0 | 0 | 0 | 0 | 4.8 | 4.7 | 11.4 | 9.3 | 14 | 14 | 9.7 | 11.6 | 11.7 | 23.3 | 26.4 | 32.7 | 32.8 | 23.1 |
| $v_2-F_1-T_1$ | 4.9 | 4.7 | 4.3 | 4.4 | 4.3 | 4.4 | 4.5 | 9.6 | 9 | 15.3 | 13.2 | 17.5 | 17.8 | 13.7 | 15.9 | 15.7 | 26.3 | 29.8 | 35.5 | 35.7 | 26.5 |
| $v_3-F_1-T_1$ | 7.4 | 6.5 | 6.1 | 5.8 | 6.5 | 6.6 | 6.5 | 11.6 | 11 | 16.6 | 14.9 | 19.6 | 19.7 | 15.6 | 18.1 | 17.3 | 27.6 | 31.2 | 36.9 | 36.9 | 28 |
| $v_1-F_2-T_1$ | −4.7 | −5.1 | −14.6 | −12.3 | −1 | −1.2 | −6.5 | 0.9 | 0.3 | 1.2 | 0 | 4.1 | 4.5 | 1.8 | 8.4 | 8.2 | 17.2 | 21.2 | 23.8 | 24.3 | 17.2 |
| $v_2-F_2-T_1$ | 0.4 | −0.3 | −9.5 | −7.2 | 3.3 | 3.1 | −1.7 | 5.3 | 5 | 5.6 | 4.4 | 7.9 | 8.3 | 6.1 | 12.5 | 12.5 | 20.8 | 23.8 | 27.5 | 27.5 | 20.8 |
| $v_3-F_2-T_1$ | 2.7 | 1.9 | −6.8 | −5.1 | 5.4 | 5.2 | 0.6 | 7.7 | 7.2 | 7.7 | 6.5 | 10.5 | 10.1 | 8.3 | 14.8 | 14.5 | 22.6 | 25.5 | 28.8 | 29.1 | 22.6 |
| $v_1-F_1-T_2$ | 2.8 | 1.4 | 0.4 | 0 | 0.9 | 0.8 | 1.1 | 7.4 | 6.7 | 11.3 | 9.4 | 14.3 | 14.4 | 10.6 | 13.9 | 12.9 | 23.3 | 26.9 | 33.5 | 33.1 | 23.9 |
| $v_2-F_1-T_2$ | 6.9 | 5.9 | 4.4 | 4.2 | 5 | 5 | 5.2 | 11.3 | 10.7 | 14.9 | 13.4 | 18 | 18.1 | 14.4 | 17.5 | 16.8 | 26.5 | 30.1 | 35.9 | 35.8 | 27.1 |
| $v_3-F_1-T_2$ | 9.1 | 8.2 | 6.5 | 6.3 | 7 | 7.1 | 7.4 | 13.5 | 13.3 | 16.9 | 15.2 | 19.9 | 19.9 | 16.5 | 19.4 | 18.7 | 28.1 | 31.7 | 37.6 | 37.4 | 28.8 |
| $v_1-F_2-T_2$ | −3.4 | −4.2 | −14.5 | −12.2 | −0.7 | −0.9 | −6 | 1.7 | 1.5 | 1.5 | 0.4 | 4.2 | 3.7 | 2.2 | 9.7 | 9.3 | 17.4 | 20.6 | 23.8 | 23.8 | 17.4 |
| $v_2-F_2-T_2$ | 1.5 | 0.3 | −9.6 | −7.3 | 3.7 | 3.5 | −1.3 | 6.3 | 5.8 | 5.7 | 4.7 | 7.8 | 8.1 | 6.4 | 13.7 | 13.2 | 21 | 24.6 | 27.2 | 26.8 | 21.1 |
| $v_3-F_2-T_2$ | 3.5 | 2.6 | −7.1 | −5 | 5.8 | 5.7 | 0.9 | 8.4 | 7.9 | 7.7 | 6.7 | 9.8 | 9.9 | 8.4 | 15.7 | 15 | 22.7 | 26 | 29.2 | 28.7 | 22.9 |

**Table A6.** Average reduction of T2 vs. T1 in same conditions.

| Condition | $D_1$ | | | | | | Avg. R (%) | $D_2$ | | | | | | Avg. R (%) | $D_3$ | | | | | | Avg. R (%) |
|---|---|---|---|---|---|---|---|---|---|---|---|---|---|---|---|---|---|---|---|---|---|
| | R1 | R2 | C1 | C2 | RC1 | RC2 | | R1 | R2 | C1 | C2 | RC1 | RC2 | | R1 | R2 | C1 | C2 | RC1 | RC2 | |
| $v_1-F_1$ | 2.8 | 1.4 | 0.4 | 0 | 0.9 | 0.8 | 1.1 | 2.6 | 2 | −0.1 | 0.1 | 0.3 | 0.4 | 0.9 | 2.3 | 1.2 | 0 | 0.5 | 0.8 | 0.3 | 0.8 |
| $v_2-F_1$ | 2 | 1.2 | 0.1 | −0.2 | 0.7 | 0.6 | 0.7 | 1.7 | 1.7 | −0.4 | 0.2 | 0.5 | 0.3 | 0.7 | 1.6 | 1.1 | 0.2 | 0.3 | 0.4 | 0.1 | 0.6 |
| $v_3-F_1$ | 1.7 | 1.7 | 0.4 | 0.5 | 0.5 | 0.5 | 0.9 | 1.9 | 2.3 | 0.3 | 0.3 | 0.3 | 0.2 | 0.9 | 1.3 | 1.4 | 0.5 | 0.5 | 0.7 | 0.5 | 0.8 |
| $v_1-F_2$ | 1.3 | 0.9 | 0.1 | 0.1 | 0.3 | 0.3 | 0.5 | 0.8 | 1.2 | 0.3 | 0.4 | 0.1 | −0.8 | 0.4 | 1.3 | 1.1 | 0.2 | −0.6 | 0 | −0.5 | 0.2 |
| $v_2-F_2$ | 1.1 | 0.6 | −0.1 | −0.1 | 0.4 | 0.4 | 0.4 | 1 | 0.8 | 0.1 | 0.3 | −0.1 | −0.2 | 0.3 | 1.2 | 0.7 | 0.2 | 0.8 | −0.3 | −0.7 | 0.3 |
| $v_3-F_2$ | 0.8 | 0.7 | −0.3 | 0.1 | 0.4 | 0.5 | 0.3 | 0.7 | 0.7 | 0 | 0.2 | −0.7 | −0.2 | 0.1 | 0.9 | 0.5 | 0.1 | 0.5 | 0.4 | −0.4 | 0.3 |

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
