# Peer review of "Reducing Carbon Emissions for the Vehicle Routing Problem by Utilizing Multiple Depots"

_sustainability, doi:10.3390/su14031264_

Round 1

Reviewer 1 Report

ID: sustainability-1532834-peer-review-v1

Title: Reducing carbon emissions for the vehicle routing problem by utilizing multiple depots

The focus of the paper is emission control by investigation of a multi-depot mode in vehicle routing. Authors proposed a branch-and-price (BAP) algorithm. Finally, a sensitivity analysis considering factors such as the number of depots, vehicle speed, customer demand, and service time is presented.

The topic may be interest of optimization researchers as it may have applications in logistic problems. However, I have some major issues about content of the paper. Please see the following comments:

- My main issue with this paper is that some terms have not defined properly. For example, “factor analysis” is defined in keywords list and later. As I know, “sensitivity analysis” is a popular approach to study how the change in the output of a mathematical model or system (numerical or otherwise) can be divided and allocated to different sources of change in its inputs.

The current paper is named it factor analysis, but sensitivity analysis is very common in the literature and so I believe that terms like sensitivity analysis better fit for replacing with the term mentioned in the paper.

-Another major issue with this paper is the format of figures. I doubt if all of them have a proper format. For example, Figure 3 does not have an informative format.  “The grouped bar chart” is a better option for this case to have 4 groups, each with 3 conditions: Group 1 V1-F1-T1, V2-F1-T1, V3-F1-T1, Group 2 V1-F2-T1, V2-F2-T1, V3-F2-T1, Group 3 V1-F1-T2, V2-F1-T2, V3-F1-T2, Group 4 V1-F2-T2, V2-F2-T2, V3-F2-T2. One bar cluster is plotted for each group, and in each cluster, one bar for each representative. Colors and positions are consistent within each cluster: for example, we should see that V1 is always in blue and plotted first, etc.

Author can also check individual performances through a grouped bar chart. There is the same problem about Figure 4.

- There is an issue about the referencing. The reference number should be immediately after the name. For example, in Page 2, the sentence “Andrea et al. proposed a branch-and-cut-and-price … heterogeneous VRPTW [17].” should be replaced with “Andrea et al. [17] proposed a branch-and-cut-and-price … heterogeneous VRPTW.”. This error has occurred in many places.

Furthermore, authors should always referee to the family names, not first names. So, Andrea et al. [17] should be Bettinelli et al. [17]. Please correct the issue for all references.

- The literature review is not broad enough. Authors should briefly mention the similar (to VRP) routing problems such as TSP and its applications to this logistic. Recent studies should be briefly added to the discussion with following citations. [a] A branch-and-cut algorithm for a traveling salesman problem with pickup and delivery. Discrete Applied Mathematics, vol. 145, pp. 126–139 [b] A transformation technique for the clustered generalized traveling salesman problem with applications to logistics, European Journal of Operational Research, vol. 285, no.2, pp. 444-457

- The Equation 6 has two min, but I wonder why just one min is not enough as the equation finally reports the min of all elements.

-In Page 9, there are two Tables 3. This is very confusing. I assume the second one should be Table 4.

- Finally, the paper benefit from a slight proofread.

Reviewer 2 Report

Title : Reducing carbon emissions for the vehicle routing problem by utilizing multiple depots
Authors : Sihan Wang, Yang Yu, Min Huang and Wei Sun
-----------------------
In this paper, the authors present a method to reduce CO2 emissions by replacing the single-depot mode with a multi-depot mode in vehicle routing. More, they identify the situations under which multiple depots could be used to effectively reduce carbon emissions.

Strengths:
. The abstract presents correctly and synthetically the paper.  
. The reference to Solomon's benchmark in the text is very positive, for the reproductibility of the results by other researchers.  
. The results (discussion) section is correctly described, well illustrated and clearly commented.  
. Conclusion section is clear and synthetic.

Weaknesses:
. The Branch and Price procedure could be more detailed.

Originality / Novelty:
. This question is well defined, not original but the way proposed to tackle this problem is original.

Significance:
. The scientific content of this paper is correct for me and deserves to be published.  
. The hypotheses are correctly identified as such, in the modelling section.
. The presented results are significant and appropriately presented.
. The technical quality of this paper is correct for me.  
. The conclusion is correctly justified and supported by the results.  
. The limits of the results obtained in this paper are not mentioned. This point could be investigated, and could easily begin with the limits of the model.  
. I took interest and pleasure to read this paper.

Quality of presentation:
. The abstract is clear and presents correctly the subject addressed in this paper.  
. This paper contains the basic sections of a scientific paper.  
. The subheadings used for the redaction of this paper make it clear.  
. This paper is clear, easy to follow and to read, and logically written.  
. The data and analyses are appropriately presented.  
. The conclusion is argumented and clear enough.

Scientific soundness:
. The subject addressed in this paper is relevant.  
. The study has been correctly designed, and is technically sound.  
. In my opinion, the analyses of the results are convincing.  
. The data presented in this study seems robust enough to draw the conclusions.  
. The methods and software are described with sufficient details to allow another researcher to reproduce the results. This point is very important, and thus very positive for the authors.

Overall evaluation:
. The authors have addressed an already studied question, with smart experiments as well as a correct bibliography. 
. This work provides an advance towards the current knowledge, clearly highlighted in the abstract.  
. The English language quality and style of this paper are appropriate and understandable.
. In my opinion, this paper is very interesting and deserves to attract a wide readership, beyond the limits of the journal's readership. I think there is an overall benefit to publish this work.
As a conclusion, my suggestion to the editor is to accept this paper for publication in Sustainability.

References :
--------------
. 39 research references, without non-research reference, out of which 2 self-references, giving an acceptable self-reference ratio equal to 5%.
. Avoid citing groups of references: [8-10], [11-13], [20-22], [24-28]. Or if you do it anyway, please comment more any of these references. 
. The bibliography of this paper is mainly composed of recent references: 1 of them are more than 10 years old, and 38 of them are less than 10 years.

Typos / Comments / Remarks:
------------------------------------
. Lack of coherence between equations (7) and (8).
. Line 329: Table 3 --> Table 4
. Line 426: "Third bullet" can be erased.

Reviewer 3 Report

The paper "Reducing carbon emissions for the vehicle routing problem by utilizing multiple depots” is interesting for journal readers. Kindly take note of the following specific comments to make it better.

I have thoroughly checked the paper, and found that this is an interesting paper which has potential towards acceptance but it needs some changes in it. 

  • The purpose, importance, and contribution of the paper must be emphasized more clearly.
  • Generally, the work is well-written and can adequately contribute to the existing literature but it needs to be edited. The literature review may be more up to date.
  • # Should use Literature Review title instead of Related work (Section 2)
  • # Some part of literature and intro is badly written. In addition, should add missing & recent literature… To be redesign again. Please visit
  • https://doi.org/10.1007/s11356-021-12637-y
  • https://doi.org/10.1007/s11356-021-16720-2
  • https://doi.org/10.1007/s11356-021-12993-9
  • So many formulas, please take out from text or give some of them in APPEX, methodology part is hard to understand…Most of them are book knowledge…please also change the name of the this part to Methodology, too
  • Finally the manuscript has some typos; the author is requested to have it edit and improve quality of the paper.
  • Discussion is ok but Conclusion is weak and Need clear future recommendation/implementation
  • Will be looking forward to seeing your reviewed manuscript.

Round 2

Reviewer 1 Report

I have read the paper once more. My comments are resolved in the current revision. The paper has a good discussion about carbon reduction in VRP. 
This can be helpful for researchers in the field.
So, it can be accepted as it is. 

Reviewer 3 Report

Based on the my suggestions a decision about this paper: Accept